# Characteristics and Expression Analysis of the MYB-Related Subfamily Gene in *Rosa chinensis*

**DOI:** 10.3390/ijms252312854

**Published:** 2024-11-29

**Authors:** Yongjie Zhu, Yuzheng Deng, Yandong Yao, Kangding Yao, Xuejuan Pan, Xuetong Wu, Zhiya Liu, Jitao Zhang, Wanyi Su, Weibiao Liao

**Affiliations:** College of Horticulture, Gansu Agricultural University, Lanzhou 730070, China; 18793284582@163.com (Y.Z.); dengyz0830@126.com (Y.D.); yyd614636237@163.com (Y.Y.); 19119882925@163.com (K.Y.); panxj@st.gsau.edu.cn (X.P.); wuxt@st.gsau.edu.cn (X.W.); lzyaaa1127@163.com (Z.L.); zjt117769@163.com (J.Z.); s18294911904@163.com (W.S.)

**Keywords:** *Rosa chinensis*, MYB-related, plant growth regulator, bioinformatics analysis, gene expression

## Abstract

MYB-related transcription factors (TFs) subfamily is a subfamily of MYB TFs, which are mainly involved in plant secondary metabolism, growth and development, and stress response. To explore the function of MYB-related subfamily genes in *Rosa chinensis*, this study systematically analyzed characters of the MYB-related subfamily members in *R*. *chinensis* with bioinformatic analysis using the genomic data of *R*. *chinensis* and investigated their expression characteristics using quantitative real-time PCR (qRT-PCR). The results show that 100 MYB-related proteins were identified in *R*. *chinensis*. Proteins are mainly found in the nucleus. Chromosome localization revealed that all MYB-related genes are mapped to seven chromosomes and are distributed in clusters. Collinear analysis shows that 13 pairs of MYB-related genes had a collinear relationship, indicating *R. chinensis* may have evolved its MYB-related subfamily gene through fragment duplication. The analysis of motifs and conserved domains shows that Motif 3 is the most conserved motif. There are numerous ABA and MeJA response elements in MYB-related genes. ABA and MeJA treatments significantly shortened the vase life of *R. chinensis*, while the flower diameter on day 3 was the largest, suggesting that ABA and MeJA might induce MYB-related gene expression during cut flower senescence. The expression of MYB-related genes is tissue specific, most of which show the highest expression levels in petals. Notably, among six plant growth regulator treatments, ABA treatment significantly increased *RcMYB002* expression in *R. chinensis*, suggesting that *RcMYB002* may be a crucial gene for ABA response. This study provides a reference for further research on the function of MYB-related genes in *R*. *chinensis*.

## 1. Introduction

Every eukaryote contains a large, functionally diverse class of transcription factors (TFs) called the MYB TFs [1]. Tobacco plants overexpressing *EgMYB2* produced thicker secondary cell walls and increased lignin content [2]. Theanine synthesis has been shown to be controlled by MYB TFs in various organs and tissues of tea plants [3]. In Arabidopsis, *MYB12* tightly regulated flavonol levels [4]. *MdMYB1* regulated anthocyanin transport and synthesis by interacting with TF *MdGSTF6* in apple [5]. Furthermore, *MdMYB6* inhibited the synthesis of anthocyanins by lowering glucose and galactose levels [6]. The R2R3-MYB-type transcriptional activator WP1 directly influenced the expression of carotenoid biosynthesis genes, such as *MtLYCb*, via its C-terminal acidic activation motif [7]. Anthocyanin synthesis required the interaction between TFs MYB and WD40. It has been shown that sweet cherry *PacMYBA* could bind to the promoters of *DFR*, *ANS*, and *UFGT* after interacting with several members of bHLH [8]. The pigmentation in sugar beet was inhibited when *BvMYB1* was silenced [9].

In MYB TFs, one or more MYB domains are present. The MYB domain consists of three helices, with the two C-terminal helices forming helix-turn-helix (HTH) structures [10]. The HTH structure recognizes the C/TAACG/TG motif, thereby conferring specificity to the MYB proteins [11]. The N-terminal helix contains a DNA-binding region composed of several unique R structures, each approximately 51 amino acids in length. The C-terminal transcriptional regulatory region exhibits significant variability, which may account for the diverse functional roles of MYB TFs [12]. In the MYB DNA-binding domain, MYB proteins are classified according to their number and arrangement of MYB repeats (R), which plays a crucial role in DNA binding and protein–protein interactions. These categories include 1R-MYBs (also referred to as MYB-related), 2R (R2R3)-MYBs, 3R (R1R2R3)-MYBs, and 4R-MYBs, corresponding to proteins containing one, two, three, and four MYB repeat, respectively. In all plant species, these four groups can be found. In plants, 2R-MYBs are essential for a number of specific processes. Most eukaryotic organisms contain members of the 3R-MYB subfamily, which regulates the cell cycle. The 4R-MYBs are the smallest subfamily of the MYBs [13,14]. Until now, MYB-related proteins have been less well studied, although they contain MYB repeats or partial MYB repeats and are only second in size to 2R-MYBs. A transcriptional activator for MYB-related TFs has been identified for the first time in *Solanum tuberosum* and was named *MybSt1* [15]. Numerous MYB-related subfamily genes have been discovered in a wide range of species, and they play a key role in physiological processes, such as circadian rhythms, secondary metabolism, and cellular development [14,15,16].

*Rosa chinensis* is a shrub belonging to the Rosaceae family that grows low and evergreen and is also a popular ornamental plant known for its rich color and aroma [17,18,19]. Poplars and grapevines have also been found to contain the MYB-related subfamily genes in recent years [20,21], but these studies are much less comprehensive than those conducted on herbaceous plants. In addition, the functional diversity of MYB-related subfamily in *R*. *chinensis* has been less studied at the molecular level. Thus, here, bioinformatics was used to identify the MYB-related subfamily in the *R*. *chinensis* genome, and their physicochemical properties, subcellular localization, chromosomal localization, collinearity, cis-acting elements, and Motifs were analyzed. Following different plant growth regulator treatments, the levels of gene expression were further determined. This study could aid future research on the mechanisms by which MYB-related subfamily genes respond to phytohormones in *R*. *chinensis*.

## 2. Results

### 2.1. Rosa chinensis MYB-Related Gene Subfamily Identification and Physicochemical Properties

Based on the *R*. *chinensis* genome data, Pfam, NCBI-CDD, and SMART online tools were used for screening MYB-related subfamily genes. Finally, 100 MYB-related subfamily members were identified. They are named RcMYB001 through RcMYB100 (Table 1). Then, ExPASy and WoLFPSORT online tools were used to analyze the physicochemical properties and subcellular localization of MYB-related subfamily members (Table 1). We found that the amino acid length of MYB-related proteins in *R*. *chinensis* varies greatly, ranging from 70 (RcMYB042) to 2010 (RcMYB095). The molecular weight (Da) is between 7965.85 (RcMYB042) and 218,668.55 (RcMYB095). By predicting its physical and chemical properties, we found that basic proteins are the main part, accounting for 62% (PI > 7). All MYB-related proteins of *R*. *chinensis* are hydrophilic proteins. It has a Grand Average of Hydropathicity (GRAVY) of around −0.737. In addition, the instability index is between 31.47 (RcMYB040) and 99.91 (RcMYB046). There are eight proteins with instability index <40, which are presumed to be unstable proteins. The rest are stable proteins, accounting for 92%. The findings suggest that MYB-related proteins might perform distinct biological roles depending on the microenvironment.

Using WoLFPSORT [https://wolfpsort.hgc.jp/ (accessed on 6 June 2024)], the subcellular localization of MYB-related protein was predicted in *R*. *chinensis*. According to Table 1, 83 MYB-related proteins were localized in the nucleus, suggesting that most of them regulate downstream genes through TFs. Whereas a few MYB-related proteins were also localized in the cytoplasm (including RcMYB030, RcMYB078 and RcMYB091), mitochondria (including RcMYB022, RcMYB023, and RcMYB066), chloroplasts (including RcMYB017, RcMYB018, RcMYB027, RcMYB047, RcMYB085, RcMYB086, and RcMYB087), and peroxisomes (including RcMYB061, RcMYB063, RcMYB064, and RcMYB092). According to the results, different MYB-related subfamily members may be involved in different physiological activities involved in the growth and development of *R*. *chinensis*.

### 2.2. Analysis of the Chromosomal Location of MYB-Related Subfamily in Rosa chinensis

MYB-related subfamily members were mapped on chromosomes 1 to 7 in *R*. *chinensis* (Figure 1). There is an uneven distribution of genes on each chromosome, with most genes being located at the top or bottom of chromosomes 1, 2, 5, 6, and 7. Genes on chromosome 4 are largely distributed at the lower end, while genes on the remaining chromosomes are more evenly distributed. There are 22 genes in the MYB-related subfamily on chromosome 3, representing 22% of the total number of genes. A total of 19 and 16 members of the MYB-related subfamily are found on chromosomes 2 and 6, 12 members are found on chromosome 5, and 11 members are found on chromosomes 1 and 4. Among the genes on chromosome 7, only nine belong to the MYB-related subfamily, which represents only 9%. In addition, most of the genes of this subfamily on Chr1, Chr2, Chr3, Chr4, Chr5, Chr6, and Chr7 are distributed in clusters, which might be related to the expansion of this subfamily of genes as well as functional differentiation.

### 2.3. Analysis of the Collinearity and Evolutionary Selection Pressure of MYB-Related Subfamily Members in Rosa chinensis

We have identified 13 pairs of replicated genes in the MYB-related subfamily (Figure 2A). The 13 pairs of genes occur both intrasegmentally and interchromosomally. Three of the MYB-related genes are replicated more than once (*RcMYB084*, *RcMYB098*, and *RcMYB100*). This indicates that the members of the MYB-related subfamily have chromosome fragment replication events during evolution, so it can be inferred that the pairs of genes may have very similar functions.

Based on the Ka/Ks values, it is possible to determine how frequently nonsynonymous mutations occur in comparison to synonymous mutations over time, providing evidence of evolutionary selection pressure on genes. Here, all MYB-related genes have Ka/Ks values less than 1 (Figure 2B). This implies that purifying selection is primarily responsible for the evolution of MYB-related genes in *R*. *chinensis*.

### 2.4. Motif Analysis of Rosa chinensis MYB-Related Subfamily

As shown in Figure 3B, 15 different motifs were identified among 100 MYB-related proteins. Motif 3 was found to have the highest frequency as all members except RcMYB001, RcMYB029, RcMYB039, RcMYB046, RcMYB079, and RcMYB080 contain Motif 3. Motifs identified in different branches display specific evolutionary distinctions, while protein sequences from the MYB-related subfamily within the same branch tend to share similar types of motifs. It was found that Motif 3 has the most primitive structure and the most conserved function in the evolution of the MYB-related subfamily. It was speculated that other conserved motifs evolve on this basis. This may allow the *R*. *chinensis* MYB-related subfamily to be more functionally diverse.

### 2.5. Cis-Acting Elements in the Rosa chinensis MYB-Related Subfamily

To analyze the potential biological function of *RcMYB-related* genes, the Plant CARE online tool was used to identify cis-acting elements in the 2000 bp upstream region of the gene initiation codon (ATG). The *R*. *chinensis* promoter region was found to contain nine cis-elements (ABRE, CAT-box, CGTCA-motif, GARE-motif, P-box, TATC-box, TCA-element, TGACG-motif, and TGA-element) related to hormone response (Figure 3C and Figure 4). Compared to other types of elements, ABRE elements are the most abundant and distributed in the majority of promoter regions, accounting for 35.6% of the total number. Approximately 86% of the family members’ promoter regions include ABRE elements, while around 73% contain methyl jasmonate-related elements. In terms of individual gene family members, the *RcMYB081*, *RcMYB082*, and *RcMYB083* promoters have only abscisic acid (ABA) responsive elements (ABRE) and may not be subject to the actions of several other classes of hormone response elements. *RcMYB002*, *RcMYB003*, *RcMYB011*, *RcMYB020*, *RcMYB021*, and *RcMYB079* are enriched in ABRE elements. The expression of these genes may contribute to the response to abscisic acid (Table 2). The presence of TCA-element elements on genes such as *RcMYB012* and *RcMYB099* suggests that these genes may be associated with salicylic acid response. Genes such as *RcMYB097* and *RcMYB098* harbor TGA elements, indicating their potential involvement in auxin-responsive regulation. Genes *RcMYB007*, *RcMYB008*, and *RcMYB009* contain CGTCA-motif and TGACG-motif elements, indicating that they may be involved in methyl jasmonate (MeJA) response. *RcMYB032* and *RcMYB036* exhibit GARE-motif and P-box elements within their promoter regions, while *RcMYB042* and *RcMYB092* include TATC-box elements in their promoters. This suggests that gibberellin signaling may regulate these genes. The genes *RcMYB049* and *RcMYB090* contain CAT-box elements on their promoter elements, suggesting that these genes may respond to phloem organization-related expression.

In the analysis of cis-acting elements, we found that they are associated with six phytohormones: ABA, 6-benzylaminopurine (6-BA), MeJA, salicylic acid (SA), gibberellic acid (GA3), and indole-3-acetic acid (IAA). Thus, cut *R*. *chinensis* was treated with six plant growth regulators. As shown in Figure 5A, *R. chinensis* treated with ABA and MeJA had a shorter vase life than the control. *R. chinensis* treated with SA had significantly longer vase life than controls. As indicated in Figure 5B, after 3 days of treatment, the diameters of flowers subjected to ABA, MeJA, and GA3 were all significantly greater than those of the control. On day 6 of treatment, the histogram reveals that the flower diameter in the 6-BA, GA3, and SA treatments exceeded that of the control, whereas the flower diameters in the IAA treatment were found to be lower than those of the control. On day 9, the histogram reveals that the flower diameter in the 6-BA, GA3, and SA treatments exceeded that of the control.

### 2.6. Analysis of MYB-Related Gene Expression in a Tissue-Specific Manner in Rosa chinensis

Based on the quantity of abscisic acid response elements present in the *R*. *chinensis* genome, the six MYB-related genes were screened for their expression specificity, as shown in Figure 4. These genes include *RcMYB002*, *RcMYB003*, *RcMYB011*, *RcMYB020*, *RcMYB021*, and *RcMYB079*. *R. chinensis* tissues were evaluated using qRT-PCR for the expression of MYB-related genes (Figure 6). Among the most highly expressed genes, *RcMYB002*, *RcMYB003*, *RcMYB011*, *RcMYB020*, and *RcMYB079* were found in petal tissues, while *RcMYB021* was found in leaves. Furthermore, the expression levels in the stem, pistil, stamen, calyx, receptacle, and prickle tissues were generally low. Thus, MYB-related genes associated with abscisic acid cis-acting elements may play an important role mainly in petals.

### 2.7. Effects of Different Hormones on MYB-Related Gene Expression in Rosa chinensis

In order to explore the role of MYB-related genes in *R*. *chinensis* petal tissue under hormonal treatments (ABA, 6-BA, MeJA, GA3, SA, and IAA), we analyzed the expression levels of six MYB-related genes. A significant upregulation of *RcMYB020* and *RcMYB079* was observed after 6 h of SA treatment compared to the control group, while significant inhibition of *RcMYB003* was observed. After 12 h of treatment with SA, *RcMYB002*, *RcMYB011*, *RcMYB020*, and *RcMYB079* demonstrated significant upregulation in expression, while *RcMYB003* exhibited a notable reduction. All members except *RcMYB003* were significantly upregulated following 24 h of SA treatment. In comparison to the control, the treatment with MeJA for 6 h resulted in a significant upregulation of the expression levels of *RcMYB020*, *RcMYB021*, and *RcMYB079*. *RcMYB021* displayed the highest expression level between 12 and 24 h after MeJA treatment. After 6 h of ABA treatment, there was a notable reduction in the expression levels of *RcMYB020* and *RcMYB021*. After 12 to 24 h of ABA treatment, all members’ expression levels increased significantly, with *RcMYB002* showing a 7.80-fold increase over the control group.

There was a significant up-regulation of *RcMYB011*, *RcMYB020*, and *RcMYB079* expression levels following 6-BA treatment for 6 h. After treatment with 6-BA for 12 to 24 h, *RcMYB011* peaked at 12 h, while the expression levels of the other genes progressively increased. In contrast, all genes subjected to IAA treatment exhibited a significant downregulation after 6 h. However, their expression levels were subsequently upregulated between 12 and 24 h. After 12 h of GA3 treatment, all members exhibited significant upregulation, with *RcMYB079* demonstrating the highest expression level. Notably, this gene maintained its peak expression even after 24 h of GA3 treatment.

## 3. Discussion

The MYB TF family is a widely distributed class of TFs in eukaryotes, and their members play important roles in growth and development, physiological metabolism, and stress response [22]. Among them, MYB-related TFs have large numbers and important roles in the plant kingdom, and their numbers have been supplemented in recent years. Research on the MYB-related subfamily across the genomes of significant terrestrial plants, as well as mosses, cycads, green algae, and red algae, found that lower plants have only a limited number of MYB-related genes [23]. Angiosperms have been found to contain a considerable amount of MYB-related gene sequences, such as 70 genes in the *Oryza sativa* genome and 68 genes in *Arabidopsis* [23,24]. In this study, 100 MYB-related genes were identified in *R*. *chinensis* (Table 1). Compared with the protein sequence of maize, which ranges from 200 (*ZmMYB167*) to 406 (*ZmMYB101*) [25], the protein sequence of *R*. *chinensis* varies significantly (Table 1), indicating that the MYB-related gene may produce more diversified functions in *R*. *chinensis*. Most of the proteins of MYB-related subfamily members of *R*. *chinensis* were located in the nucleus (Table 1), which is similar to that of *Zea mays*, *Solanum tuberosum*, and *Osmanthus fragrans* [25,26,27]. Analysis of conserved motifs of the MYB-related subfamily found that Motif 3 is the most conserved, but it is noteworthy that six members do not contain this Motif 3 (Figure 3). We speculate that these members may no longer need the old, conserved motifs as they develop new functions [28]. All identified genes were mapped to seven chromosomes according to chromosome localization analysis. The chromosome distribution shows that similar to the distribution characteristics of MYB-related genes in *Capsicum annuum* [29], most family members cluster on certain fragments to form clusters, suggesting that these genes may have a common biological function.

Gene replication events lead to the diversity of family member functions, which are of great significance in plant genome evolution [30]. Different plant species have unique gene replication events in the MYB-related subfamily. For example, 103 MYB-related genes were identified in *Capsicum annuum* and randomly distributed across the chromosome, and there are six collinear gene pairs [29]. Thirteen SlMYB-related genes were identified in *Solanum lycopersicum*, including one collinear gene pair [26]. In our study, 13 MYB-related collinear gene pairs were identified, accounting for 26% of the members of the *R*. *chinensis*. Based on the analysis of the MYB-related subfamily, fragment duplication events within MYB-related genes may have played a pivotal role in *R. chinensis’* evolutionary divergence. These recurring events are crucial for plants to adapt to complex environmental conditions through gene number and functional diversity [31]. As a result of replication events, MYB-related genes are not only multiplied but also acquire new functions or become more efficient through secondary mutations and selections [32]. Wang et al. [33] demonstrated that the MYB-related gene underwent duplication and differentiation in plants, resulting in an increased number of members with diverse response functions. It facilitates the plants’ ability to cope with biotic and abiotic stresses [33]. Notably, although the replication of the MYB-related subfamily gene led to diversity in its membership and function, not all copied genes could be preserved. There may have been functional redundancies or harmful mutations that led to some duplicated genes being eliminated, while others have been retained by natural selection in order to perform new functions [34]. Therefore, in the investigation of MYB-related gene duplication events, it is essential to thoroughly consider the occurrence, retention mechanisms, and functional alterations associated with these duplications. In spite of this, more research is needed to determine the precise mechanisms and impacts associated with the replication of MYB-related subfamily genes.

By binding to trans-acting factors, cis-acting elements are the critical ‘switches’ that regulate gene expression in plants [35]. Among many plant species, researchers have found diverse cis-acting elements involved in abiotic stress response in the promoter regions of MYB-related subfamily genes, which suggests that MYB-related genes may mediate the transcriptional response of most plants to abiotic stress [36]. Therefore, identification of cis-acting elements in the promoters of MYB-related genes can help to understand gene function. This study identified nine cis-acting elements within MYB-related genes that are related to phytohormones (Figure 3). Similar to Solanum melongena [37], MeJA response elements (CGTCA-Motif and TGACG-Motif) and ABA response elements (ABRE) are found in the majority of MYB-related members in *R*. *chinensis*. In numerous studies, MYB-related genes are associated with transcriptional regulatory elements connected to abiotic stress responses [21,38]. Plant defense hormones, such as GA and ABA, can induce the *PtrMYBs* gene to improve resilience to abiotic stress, for instance [21]. In *Boehmeria nivea*, the expression of *BnMYB2* is induced by cadmium (Cd) stress, and its overexpression leads to enhanced tolerance and accumulation of Cd in Arabidopsis [38]. Therefore, *BnMYB2* positively regulates Arabidopsis tolerance and accumulation of Cd. It can enhance the efficiency of Cd removal in plants [38]. In our study, we also observed that treatment with ABA and MeJA significantly shortened the vase life of cut *R*. *chinensis* flowers (Figure 5). Therefore, we propose that ABA and MeJA could influence the expression of genes related to the MYB subfamily in *R*. *chinensis*.

Using different treatment conditions, gene expression levels can be compared to assess how cis-acting elements operate in stress response. To date, numerous studies have investigated MYB-related expressions in other plant stresses [39,40,41,42,43]. For example, Li et al. [40] found that drought stress led to a strong up-regulation of *BnMRD107* expression in *Brassica rapa* [40]. The overexpression of *BnMRD107* improved the resilience of *Brassica rapa* seedlings to osmotic stress, highlighting its beneficial role in drought response. Research revealed that *ZmRL6* exhibits high expression levels in maize when exposed to drought [43]. Furthermore, the overexpression of *ZmRL6* was found to improve drought tolerance, whereas the knockout of *ZmRL6* using CRISPR-Cas9 led to heightened sensitivity to drought [43]. Overexpression of the MYB-related gene *OsMYBR1* increased drought tolerance and decreased ABA sensitivity in *Oryza sativa*. In transgenic plants under drought stress and ABA treatment, with abiotic stress, which may have been unknown, the transcription levels of four stress-related genes were markedly elevated: *OsP5CS1*, *OsProt*, *OsLEA3*, and *OsRabl6* [42]. Salt and freezing stress tolerance may be increased in *Arabidopsis thaliana* by *AhMYB30* through the DREB/CBF and ABA pathways [39]. *Oryza sativa’s* salt tolerance was significantly improved by overexpression of *OsMYB48-1* [41]. We found that *RcMYB002*, *RcMYB003*, *RcMYB011*, *RcMYB020*, *RcMYB021*, and *RcMYB079* display the largest quantity of ABA response elements (ABRE) associated with hormone response (Figure 4). As a result, we selected these six genes to investigate MYB-related genes’ stress response in hormones. In response to ABA treatment, all members showed elevated expression levels, with *RcMYB002* increasing 7.80-fold versus control (Figure 7). Based on these results, *RcMYB002* may have a positive effect on hormone responses. However, their potential as crucial regulators of hormonal stress and their mechanisms need to be further investigated. Therefore, we can speculate that additional MYB-related genes are associated with regulatory functions.

## 4. Materials and Methods

### 4.1. The Identification of MYB-Related Gene Subfamily in Rosa chinensis and Their Physicochemical Property Analysis

The *R*. *chinensis* genome file and annotation file (GCF_002994745.2) were obtained from the NCBI online platform [https://www.ncbi.nlm.nih.gov/datasets/genome/GCF_002994745.2/ (accessed on 28 May 2024)]. *R*. *Chinensis* CDS sequences were extracted through the “Gtf/Gff3 Sequences Extract” feature of TBtools-II software (v2.136), and the software’s “Batch TranSlate CDS to Protein” function was then used to convert CDS sequences to protein sequences. Subsequently, the hidden Markov model (HMM) file of the MYB DNA-binding domain (PF00249) was downloaded from the Pfam online database [https://www.ebi.ac.uk/interpro/entry/pfam/PF00249/ (accessed on 1 June 2024)]. Then, the protein sequences in the *R*. *Chinensis* genome database were compared with the MYB hidden Markov model (HMM) using the “Simple HMM Search” function in TBtools-II (v2.136) [21]. To further determine whether an identified protein belongs to the MYB gene family, the Pfam online database [https://www.ebi.ac.uk/interpro/ (accessed on 2 June 2024)] and the “Batch CD-Search CDD Tool” function of NCBI-CDD [https://www.ncbi.nlm.nih.gov/cdd (accessed on 2 June 2024)] were used to analyze the protein domain, and those which do not contain MYB domains were deleted. Then, candidate MYB-related TFs were analyzed for SANT/MYB domains using SMART [http://smart.embl.de/ (accessed on 2 June 2024)], and proteins containing one R repeat were retained. Finally, the members of the *R*. *Chinensis* MYB-related gene subfamily were obtained, and these genes were named *RcMYB001–RcMYB100*.

The physicochemical properties of the MYB-related protein subfamily, including parameters such as amino acid length, isoelectric point (pI), molecular weight (Da), instability index, and the grand average of hydropathicity (GRAVY), were comprehensively analyzed using the “ProtParam” function of ExPASy online tool [http://web.expasy.org/protparam/ (accessed on 5 June 2024)]. Furthermore, subcellular localization predictions were performed with the WoLFPSORT platform [https://wolfpsort.hgc.jp/ (accessed on 6 June 2024)].

### 4.2. Chromosomal Localization, Collinearity Analysis, and Ka (Nonsynonymous)/Ks (Synonymous) Analysis

Using the *R*. *chinensis* genome annotation file, the MYB-related genes were mapped and visualized on chromosomes using the “Gene Location Visualize” function in TBtools-II (v2.136).

In the collinearity analysis, the “Fasta Stats” and “Table Row Extract or Filter” functions of TBtools-II (v2.136) were used to extract chromosome length and gene density information using the *R*. *chinensis* genome file and annotation file, respectively. Then, using the genome file and annotation file, we simplified and extracted the collinearity information by employing four functions in TBtools-II (v2.136): “One Step McscanX”, “File Merge For McscanX”, “Quick Run McscanX Wrapper”, and “File Transformat for MicroSynteny Viewer”. Next, the annotation file was used to extract gene position information on the chromosomes using the “GFF3/GTF Gene Position (Info.) Parse” function. Finally, the “Advanced Circos” function in TBtools-II (v2.136) was used to integrate and visualize the above information.

Through the CDS sequence and protein sequence in the *R*. *chinensis* genome and the information of gene pairs (obtained by collinear visualization), the selection and evolutionary pressure values of MYB-related genes in *R*. *chinensis* were extracted by “Simple Ka/Ks Calculator (NG)” function in TBtools-II (v2.136) [44]. Then, the visualization was performed using the “3D Scatter Plot” function in Origin (9.8.0) software.

### 4.3. An Analysis of Protein Conserved Domains

The protein sequences of *R. chinensis* MYB-related subfamily members were subjected to analysis using the MEME online program [https://meme-suite.org/meme/tools/meme (accessed on 8 June 2024)] to identify conserved motifs. Subsequently, these conserved motifs were visualized and further analyzed utilizing the ‘Gene Structure View (Advances)’ feature in TBtools-II (v2.136) software.

### 4.4. Analysis of Cis-Acting Elements in Rosa chinensis MYB-Related Genes

An analysis was conducted on a 2000-bp promoter sequence extracted from upstream of the MYB-related genes in *R*. *chinensis*. Cis-acting elements in the sequences were identified using the Plant CARE online tool [http://bioinformatics.psb.ugent.be/webtools/plantcare/html (accessed on 10 June 2024)] and visualized with TBtools-II (v2.136).

### 4.5. Plant Material, Treatments, and Parameter Determination

In this study, the fresh-cut *R*. *Chinensis* ‘movie star’ without pests and diseases was selected as the material. The length, thickness, color, and opening degree of the flower were basically the same. The bases of the flower branches were cut off at a 45° angle, the stalks were kept at 35 cm, and the top two to three compound leaves were retained. The stem bases were inserted into a bottle of water and rehydrated for 2 h, and then were put into a vase with different treatment liquids. The temperature was maintained at (20 ± 1) °C, the relative humidity was 60% ± 5%, and the light intensity was 15 µmol/m^2^/s under fluorescent lamp irradiation for 10 h (7:00–17:00) every day. The treatments include distilled water (the control), 0.5 µmol/L ABA, 60 mg/L 6-BA, 5 mg/L MeJA, 50 µmol/L GA_3_, 10 mg/L IAA, and 50 µmol/L SA, respectively [45,46,47,48,49,50]. There were three replicates per treatment and 12 flowers per replicate. The treatment fluid was changed regularly every day.

The vase life is from the first day of treatment to the number of days when petals wilt or flower heads bend. The flower diameter is defined as the maximum width of each flower and was measured with a vernier caliper. Each replicate consisted of 10 flowers, the diameter of which was measured at 0, 3, 6, and 9 d after treatment.

### 4.6. RNA Extraction and qRT-PCR Fluorescence Quantification

Stem, leaf, petal, pistil, stamen, calyx, receptacle, and prickle tissues of *R*. *chinensis* were selected as samples (0.5 g) for expression analysis of MYB-related genes in different tissues. The samples were immediately frozen in liquid nitrogen and stored at −80 °C. In addition, the expression pattern analysis in petals under different plant growth regulators at 0, 6, 12, and 24 h was conducted.

The samples were extracted with TRIzol reagent (Invitrogen, Carlsbad, CA, USA) to obtain total RNA [51,52]. Pultton P100+ ultra-micro spectrophotometers (Wuzhou Dongfang, Beijing, China) were used to determine the purity and concentration of RNA. Further experiments were conducted with RNA samples with A260/A280 ratios of 2.0 and 2.1. Afterward, cDNA was synthesized using the FastQuant First Strand cDNA Synthesis Kit (Tianen, Beijing, China). Conditions were 37 °C for 15 min, 85 °C for 5 s, and then 4 °C for the final reaction. For qRT-PCR, we used a LightCycler 480 Real-Time PCR System (Roche Applied Science, Penzberg, Germany) and SYBR Green Premix Pro Taq HS Premix kit (Hunan Aikori Biotechnology Co., Ltd., Changsha, China).

The quantitative real-time PCR (qRT-PCR) reaction mixture comprised 10 µL of 2× SYBR Green Pro Taq HS Premix, 0.4 µL of forward primer, 0.4 µL of reverse primer, 2 µL of complementary DNA (cDNA), and 7.2 µL of double-distilled water (ddH2O). The primers employed in the qRT-PCR were designed using Primer Premier 5.0 software (Premier Biosoft Corporation, San Francisco, CA, USA), with Rcactin (GenBank accession number AB239794) serving as the internal reference gene. The sequences of the above primers are listed in Appendix A. The 2^−∆∆CT^ calculation method was used to calculate the relative expression of each gene according to Schnittger and Livak [53]. Each gene’s relative expression was calculated by comparing it with that at 0 h for each treatment. All experiments were performed three times separately.

### 4.7. Statistical Analysis

The analysis of the data was conducted using SPSS version 22.0. Three biological replicates were used in all experiments. All data were gathered from three independent experiments. Duncan’s multiple range test was used to analyze statistical differences between measurements taken at different times or under different treatment conditions. At a probability level of 0.05, differences were considered significant.

## 5. Conclusions

In our study, we conducted a comprehensive identification of 100 MYB-related genes in *R*. *chinensis* and performed an in-depth analysis of their physicochemical properties, chromosomal localization, subcellular localization, conserved motifs, collinear relationships, cis-acting elements, evolutionary selection pressures, tissue-specific expression profiles, and overall expression patterns. According to cis-acting elements analysis, most MYB-related genes contain ABRE and MeJA response elements. After treatment with six plant growth regulators, it was discovered that the vase life of *R*. *chinensis* treated by ABA and MeJA was the shortest, and their flower diameter on day 3 was the largest. The analysis of tissue expression indicates that MYB-related genes linked to abscisic acid response elements showed the highest levels of expression in the petals. All six plant growth regulators induced the expression of MYB-related genes linked to ABA response elements. Notably, *RcMYB002* was highly expressed under ABA treatment, indicating that *RcMYB002* may be related to the senescence of cut *R*. *chinensis* flowers. Accordingly, the present study may suggest future research into the role of MYB-related genes in the senescence of cut *R*. *chinensis* flowers and provide a theoretical basis for understanding their functions.

## Figures and Tables

**Figure 1 ijms-25-12854-f001:**
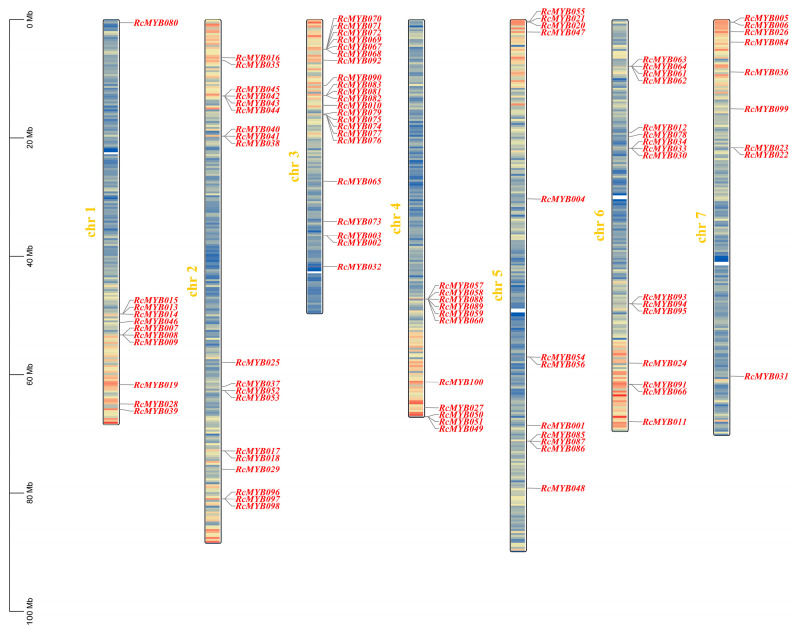
*Rosa chinensis* chromosomal localization of MYB-related genes. A chromosomal map shows the physical location of 100 *R*. *chinensis* MYB-related genes. On the left, chromosomal numbers are shown, gene names are shown in red, and a scale bar appears. Chromosomal color changes indicate the number of genes clustered.

**Figure 2 ijms-25-12854-f002:**
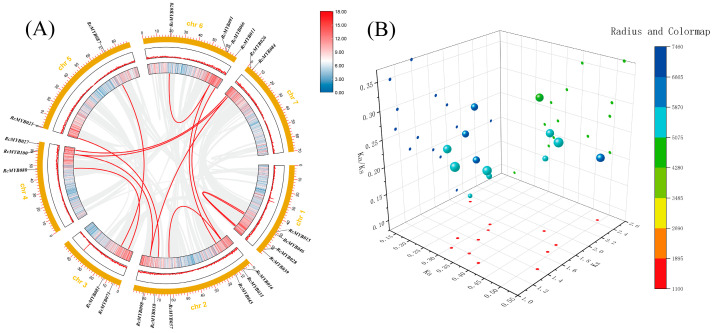
*Rosa chinensis* MYB-related subfamily collinearity and selection pressure analysis. (**A**) This figure illustrates the collinearity of MYB-related genes in *R*. *chinensis*, with the gray lines representing co-associated genes and the red lines representing co-associated genes of the MYB-related genes. (**B**) Analysis of homologous gene pairs within the MYB subfamily in terms of evolutionary selection pressure. Ka’s value is on the *X*-axis, K’s value on the *Y*-axis, and the Ka/K’s ratio on the *Z*-axis, with color scales showing fold change normalization based on log2 transformation.

**Figure 3 ijms-25-12854-f003:**
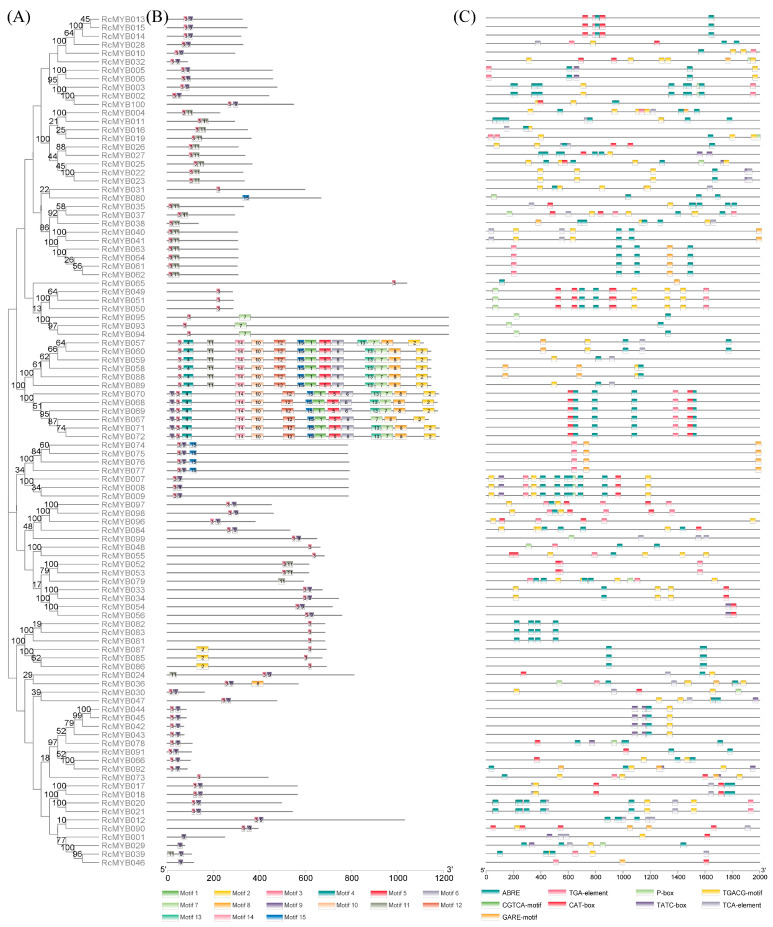
Phylogenetic tree, motif, and cis-acting element of MYB-related in *Rosa chinensis*. (**A**) Based on the MYB-related protein sequences. EGA 7.0 (7.0.26) was used to construct a phylogenetic tree using neighbor-joining. (**B**) Analysis of MYB-related subfamily motifs with conserved structural domains in *R*. *chinensis*. The numbers in the figure represent different motifs. (**C**) The MYB-related subfamily of *R*. *chinensis* contains nine cis-acting elements involved in plant hormone regulation. Each MYB-related gene is proportionally located, its length is proportional to the length of the DNA sequence, and the color represents its cis-acting elements.

**Figure 4 ijms-25-12854-f004:**
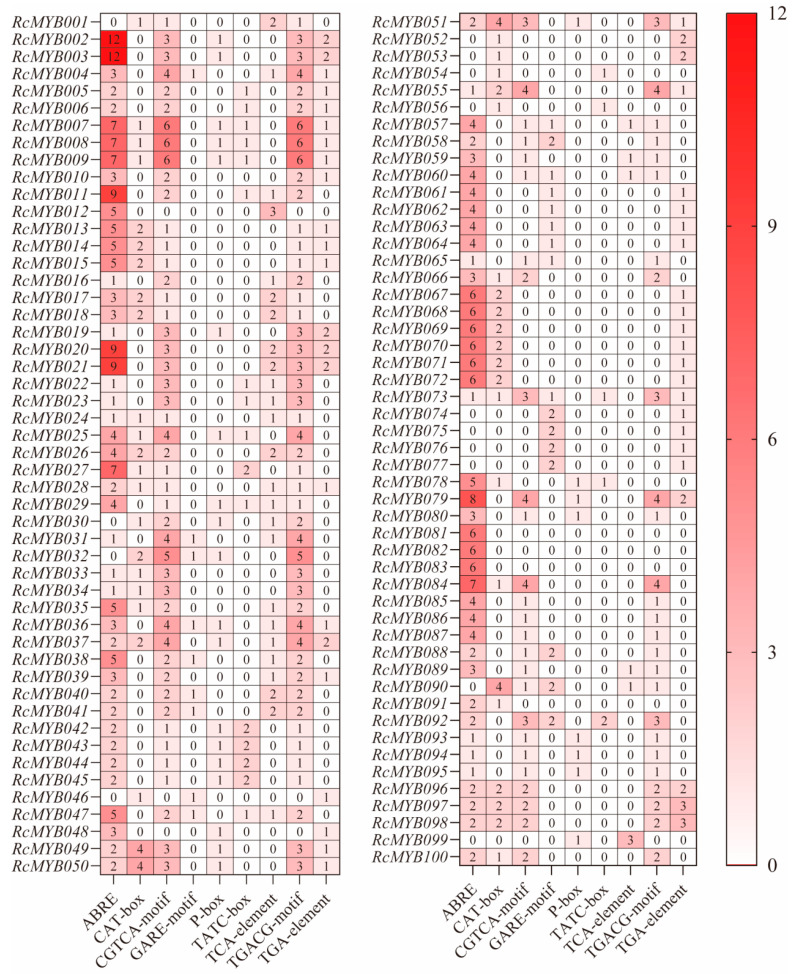
A grid representing the number of cis-acting elements in different MYB-related genes of *Rosa chinensis*. Each color in the grid represents one of the different cis-acting elements.

**Figure 5 ijms-25-12854-f005:**
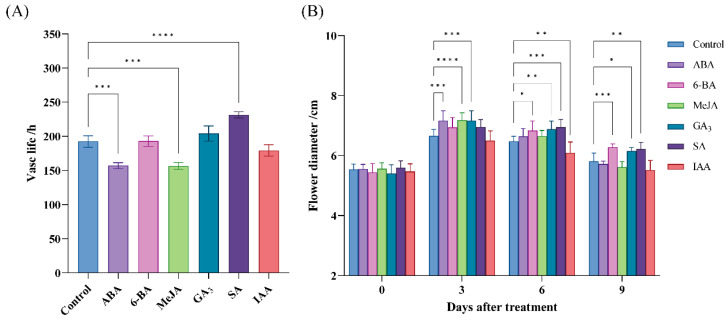
Effects of different exogenous plant hormones on senescence of *Rosa chinensis*. (**A**) Effects of exogenous phytohormones on vase life of *R*. *chinensis*. (**B**) The impact of various exogenous phytohormone treatments on the flower diameter of *R*. *chinensis*. The asterisks (*) indicate that there is a meaningful difference in expression levels between phytohormone-treated and control groups (* *p* < 0.05, ** *p* < 0.01, *** *p* < 0.001, and **** *p* < 0.0001). Reverse osmosis (RO) water-treated samples were used as controls.

**Figure 6 ijms-25-12854-f006:**
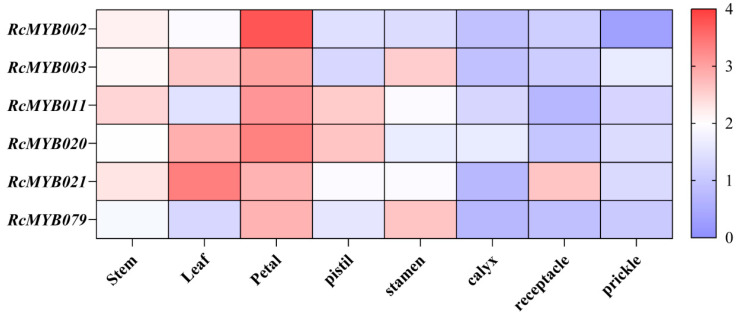
Expression of selected MYB-related genes in a tissue-specific manner in *Rosa chinensis*. The heat map shows low/medium/high expressions in blue/white/red, respectively.

**Figure 7 ijms-25-12854-f007:**
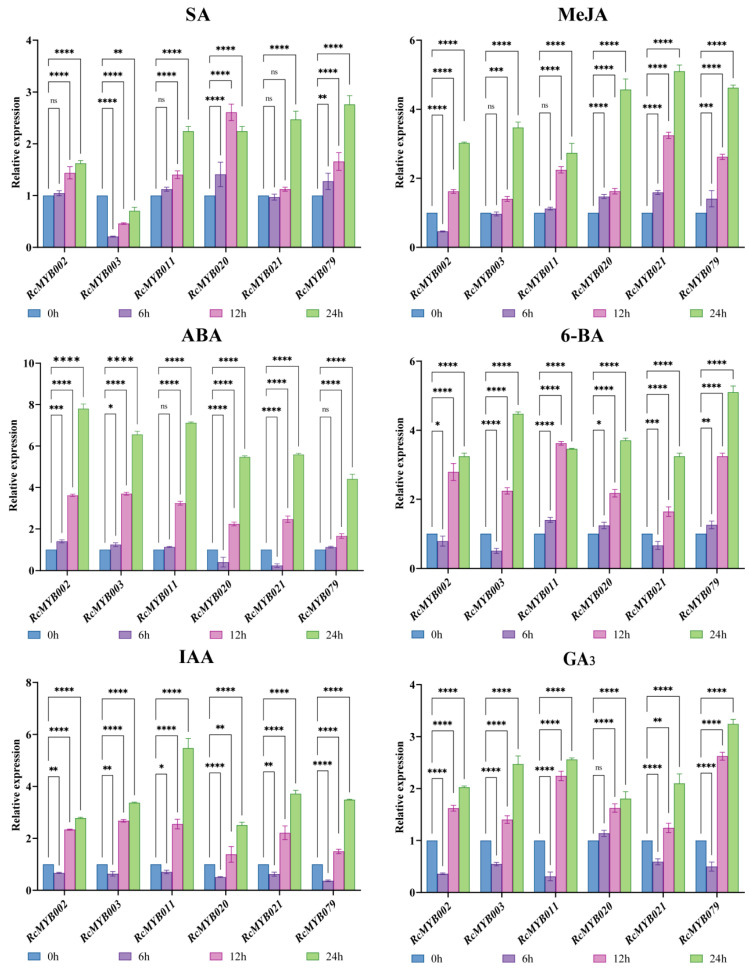
Investigating the expression of MYB-related genes in the petals of *Rosa chinensis* following different plant growth regulators. The asterisks (*) indicate that there is a meaningful difference in expression between treated and control groups (* *p* < 0.05, ** *p* < 0.01, *** *p* < 0.001, and **** *p* < 0.0001). “ns” indicates that there is no expression difference between the treatment group and the control group. 0 h-treated samples served as controls.

**Table 1 ijms-25-12854-t001:** Physicochemical properties of members of the *Rosa chinensis* MYB-related subfamily.

Gene Name	Gene ID	Amino Acid	Isoelectric Point (PI)	Molecular Weight (Da)	Instability Index	Grand Average of Hydropathicity	Subcellular Localization
*RcMYB001*	rna-XM_024303488.2	245	8.20	27,679.77	57.02	−0.925	nucl
*RcMYB002*	rna-XM_040516774.1	435	6.68	48,071.60	49.96	−0.783	nucl
*RcMYB003*	rna-XM_024333184.2	469	5.99	51,708.40	49.50	−0.792	nucl
*RcMYB004*	rna-XM_024344553.1	225	9.71	25,476.45	54.29	−0.804	nucl
*RcMYB005*	rna-XM_040510707.1	449	6.73	492,58.82	54.59	−0.772	nucl
*RcMYB006*	rna-XM_024314487.2	451	6.49	49,502.04	55.36	−0.784	nucl
*RcMYB007*	rna-XM_024320630.2	775	5.84	84,351.53	57.62	−0.757	nucl
*RcMYB008*	rna-XM_024320631.2	775	5.84	84,351.53	57.62	−0.757	nucl
*RcMYB009*	rna-XM_040517791.1	775	5.84	84,351.53	57.62	−0.757	nucl
*RcMYB010*	rna-XM_024335550.2	289	7.06	31,883.00	38.11	−0.492	nucl
*RcMYB011*	rna-XM_024308944.1	288	9.33	32,167.85	56.22	−0.691	nucl
*RcMYB012*	rna-XM_024307506.2	1016	4.78	111,028.19	55.51	−0.845	nucl
*RcMYB013*	rna-XM_040512421.1	321	9.09	35,500.98	46.47	−0.638	nucl
*RcMYB014*	rna-XM_040512422.1	314	9.01	34,731.22	44.56	−0.548	nucl
*RcMYB015*	rna-XM_024320293.2	341	8.86	37,796.62	46.83	−0.598	nucl
*RcMYB016*	rna-XM_024325719.2	343	9.59	37,208.62	58.02	−0.632	nucl
*RcMYB017*	rna-XM_040514975.1	556	6.95	62,399.95	46.28	−0.622	chlo
*RcMYB018*	rna-XM_024329759.2	557	6.95	62,487.03	47.35	−0.622	chlo
*RcMYB019*	rna-XM_024308587.2	359	9.83	38,737.73	66.61	−0.635	nucl
*RcMYB020*	rna-XM_040506100.1	489	6.04	54,955.35	52.80	−0.890	nucl
*RcMYB021*	rna-XM_024300926.2	535	6.20	60,452.89	56.27	−0.839	nucl
*RcMYB022*	rna-XM_024315383.2	324	6.22	35,079.73	59.45	−0.733	mito
*RcMYB023*	rna-XM_024315382.2	329	6.27	35,611.38	60.78	−0.706	mito
*RcMYB024*	rna-XM_024307745.2	799	5.73	86777.62	54.04	−0.566	nucl
*RcMYB025*	rna-XM_024326029.2	363	7.14	39,085.71	72.32	−0.658	nucl
*RcMYB026*	rna-XM_024317006.2	304	8.99	32,831.54	54.59	−0.610	nucl
*RcMYB027*	rna-XM_024338228.2	332	9.19	35,840.19	69.30	−0.521	chlo
*RcMYB028*	rna-XM_024320165.2	323	8.31	35,362.79	41.70	−0.593	nucl
*RcMYB029*	rna-XM_024327154.2	74	4.29	8524.14	65.71	−0.897	nucl
*RcMYB030*	rna-XM_024308285.2	159	4.78	18,465.38	45.56	−1.094	cyto
*RcMYB031*	rna-XM_024314099.2	588	8.77	62,596.12	33.14	−0.596	nucl
*RcMYB032*	rna-XM_024334815.1	86	10.03	10,106.48	34.66	−0.987	nucl
*RcMYB033*	rna-XM_040508845.1	663	5.55	72626.11	46.92	−0.839	nucl
*RcMYB034*	rna-XM_024309982.2	733	5.04	80,463.93	46.84	−0.936	nucl
*RcMYB035*	rna-XM_024325070.2	327	8.33	35,983.63	54.22	−0.700	nucl
*RcMYB036*	rna-XM_024315470.2	560	5.04	61,470.67	39.64	−0.550	nucl
*RcMYB037*	rna-XM_024325522.2	288	9.62	32,214.73	43.52	−0.625	nucl
*RcMYB038*	rna-XM_024322174.1	132	5.20	15,139.97	66.03	−0.870	nucl
*RcMYB039*	rna-XM_024321222.2	103	9.51	12,462.34	66.81	−0.897	nucl
*RcMYB040*	rna-XM_024325980.2	301	9.63	32,714.25	31.47	−0.544	nucl
*RcMYB041*	rna-XM_024325981.2	301	9.63	32,714.25	31.47	−0.544	nucl
*RcMYB042*	rna-XM_024329073.2	70	8.03	7965.85	91.79	−0.989	nucl
*RcMYB043*	rna-XM_024329072.2	71	9.05	7994.89	90.63	−0.942	nucl
*RcMYB044*	rna-XM_024329069.2	81	8.96	9285.29	80.35	−1.083	nucl
*RcMYB045*	rna-XM_024329070.2	81	8.96	9285.29	80.35	−1.083	nucl
*RcMYB046*	rna-XM_024310889.2	112	9.77	13,410.06	99.91	−1.205	nucl
*RcMYB047*	rna-XM_024301371.2	468	6.36	51,507.20	46.65	−0.522	chlo
*RcMYB048*	rna-XM_024300852.2	653	5.33	71,856.77	39.65	−0.717	nucl
*RcMYB049*	rna-XM_024338211.2	279	8.60	32,009.67	48.75	−0.671	nucl
*RcMYB050*	rna-XM_024338210.2	281	7.75	32,137.71	46.95	−0.675	nucl
*RcMYB051*	rna-XM_024338209.2	283	8.80	32,431.21	47.93	−0.654	nucl
*RcMYB052*	rna-XM_024327504.2	606	6.81	66,568.88	58.47	−0.596	nucl
*RcMYB053*	rna-XM_024327505.2	605	6.81	66,440.75	58.36	−0.591	nucl
*RcMYB054*	rna-XM_024305333.2	707	5.15	77,993.64	46.64	−0.882	nucl
*RcMYB055*	rna-XM_024305608.2	671	5.53	73,841.07	39.60	−0.671	nucl
*RcMYB056*	rna-XM_024305331.2	747	5.08	82,193.14	46.59	−0.860	nucl
*RcMYB057*	rna-XM_024339648.2	1096	9.07	123,247.94	51.88	−0.745	nucl
*RcMYB058*	rna-XM_040518381.1	1128	9.16	126,531.67	51.46	−0.740	nucl
*RcMYB059*	rna-XM_024339646.2	1128	9.11	126,518.59	51.49	−0.740	nucl
*RcMYB060*	rna-XM_024339645.2	1128	9.11	126,491.56	51.45	−0.738	nucl
*RcMYB061*	rna-XM_024311578.2	299	9.52	33,036.87	48.81	−0.571	pero
*RcMYB062*	rna-XM_040509535.1	303	9.50	33,514.37	47.85	−0.584	nucl
*RcMYB063*	rna-XM_024311575.2	303	9.52	33,532.36	49.94	−0.603	pero
*RcMYB064*	rna-XM_024311576.2	303	9.52	33,532.36	49.94	−0.603	pero
*RcMYB065*	rna-XM_024335500.2	1025	5.56	114,969.31	51.39	−0.823	nucl
*RcMYB066*	rna-XM_024311610.2	99	9.64	11,224.62	69.99	−0.827	mito
*RcMYB067*	rna-XM_024332179.2	1118	9.29	124,473.03	55.06	−0.755	nucl
*RcMYB068*	rna-XM_024332177.2	1155	9.21	128,477.53	53.98	−0.747	nucl
*RcMYB069*	rna-XM_024332175.2	1157	9.21	128,718.82	54.19	−0.745	nucl
*RcMYB070*	rna-XM_024332174.2	1161	9.21	129,192.41	54.44	−0.743	nucl
*RcMYB071*	rna-XM_024332172.2	1163	9.21	129,433.70	54.65	−0.742	nucl
*RcMYB072*	rna-XM_040516445.1	1163	9.21	129,433.70	54.65	−0.742	nucl
*RcMYB073*	rna-XM_024331881.2	432	9.41	48,959.90	53.52	−0.683	nucl
*RcMYB074*	rna-XM_024331551.2	771	7.09	85,294.11	52.32	−0.583	nucl
*RcMYB075*	rna-XM_024331550.2	772	7.09	85,365.19	52.26	−0.580	nucl
*RcMYB076*	rna-XM_024331549.2	778	7.09	85,895.77	51.93	−0.568	nucl
*RcMYB077*	rna-XM_024331548.2	779	7.09	85,966.85	52.13	−0.565	nucl
*RcMYB078*	rna-XM_024312219.2	106	9.26	11,716.95	72.64	−0.976	cyto
*RcMYB079*	rna-XM_024331915.2	582	9.09	66,370.63	61.45	−0.989	nucl
*RcMYB080*	rna-XM_040507768.1	657	9.78	73,093.15	67.12	−1.230	nucl
*RcMYB081*	rna-XM_024335784.2	674	9.23	74,653.32	44.20	−0.651	nucl
*RcMYB082*	rna-XM_024335786.2	674	9.23	74,653.32	44.20	−0.651	nucl
*RcMYB083*	rna-XM_024335788.2	674	9.23	74,653.32	44.20	−0.651	nucl
*RcMYB084*	rna-XM_024314865.2	524	4.50	58,581.92	40.48	−0.735	nucl
*RcMYB085*	rna-XM_024300794.2	662	9.01	72,970.44	54.03	−0.605	chlo
*RcMYB086*	rna-XM_024300792.2	680	9.02	74,688.37	53.54	−0.606	chlo
*RcMYB087*	rna-XM_024300793.2	680	9.02	74,688.37	53.54	−0.606	chlo
*RcMYB088*	rna-XM_040518382.1	1128	9.14	126,538.74	52.35	−0.736	nucl
*RcMYB089*	rna-XM_024339647.2	1128	9.14	126,538.74	52.35	−0.736	nucl
*RcMYB090*	rna-XM_024331150.2	388	7.96	43,391.46	43.68	−0.695	nucl
*RcMYB091*	rna-XM_024308864.2	102	9.00	11,584.77	63.90	−1.040	cyto
*RcMYB092*	rna-XM_024334178.2	85	9.10	9577.69	73.46	−0.706	pero
*RcMYB093*	rna-XM_024309508.2	1992	6.44	216,939.63	62.01	−0.839	nucl
*RcMYB094*	rna-XM_024309506.2	2010	6.39	218,668.55	61.80	−0.828	nucl
*RcMYB095*	rna-XM_024309507.2	2010	6.39	218,668.55	61.80	−0.828	nucl
*RcMYB096*	rna-XM_024325621.2	376	4.38	41,770.42	51.88	−0.631	nucl
*RcMYB097*	rna-XM_024325620.2	445	4.51	49,513.86	53.13	−0.693	nucl
*RcMYB098*	rna-XM_024325619.2	453	4.54	50,566.12	54.32	−0.693	nucl
*RcMYB099*	rna-XM_024318014.2	640	5.21	71,904.19	47.03	−0.635	nucl
*RcMYB100*	rna-XM_024338039.2	540	4.48	60,107.71	58.01	−0.758	nucl

Note: ‘nucl’, ‘chlo’, ‘mito’, ‘cyto’, and ‘pero’ stand for nucleus, chloroplast, mitochondrion, cytoplasm, and peroxisome, respectively.

**Table 2 ijms-25-12854-t002:** The roles of hormone-related cis-acting elements in the MYB-related subfamily of *Rosa chinensis*.

Cis-Element	Number of Genes	Sequence of Cis-Element	Functions of Cis-Elements
ABRE	325	ACGTG	cis-acting element involved in the abscisic acid responsiveness
CAT-box	75	GCCACT	cis-acting regulatory element related to meristem expression
CGTCA-motif	162	CGTCA	cis-acting regulatory element involved in the MeJA-responsiveness
GARE-motif	33	TCTGTTG	gibberellin-responsive element
P-box	27	CCTTTTG	gibberellin-responsive element
TATC-box	27	TATCCCA	cis-acting element involved in gibberellin-responsiveness
TCA-element	43	CCATCTTTTT	cis-acting element involved in salicylic acid responsiveness
TGACG-motif	162	TGACG	cis-acting regulatory element involved in the MeJA-responsiveness
TGA-element	60	AACGAC	auxin-responsive element

## Data Availability

All data, tables, and figures in this manuscript are original and are contained within the article and Appendix A.

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
