# Peer review of "Characteristics and Expression Analysis of the MYB-Related Subfamily Gene in Rosa chinensis"

_ijms, 2024, doi:10.3390/ijms252312854_

Round 1
Reviewer 1 Report
Comments and Suggestions for Authors
Zhu et al. identified and analyzed 100 MYB-related genes in Rosa chinensis, which are involved in various physiological functions and responses to plant hormones, notably abscisic acid (ABA) and methyl jasmonate (MeJA). These genes are predominantly localized in the nucleus, with expression levels varying across different tissues and showing the highest expression in petal tissues, particularly under ABA treatment. The findings suggest that specific MYB-related genes, such as RcMYB002, may play a crucial role in the senescence of cut Rosa chinensis flowers, laying a foundation for future research into the regulatory mechanisms governing floral development and stress responses. However, a few minor issues need to be addressed.
1- Lines 345–346: The Rosa chinensis genome file was obtained from the NCBI platform [https://www.ncbi.nlm.nih.gov/ (accessed on 28 May 2024)]. Please provide the accession number of the genome used and the direct link for easy access.
2- Lines 348–350: A Hidden Markov Model (HMM) search was performed on the R. chinensis genome database using the Pfam MYB DNA-binding domain (PF00249) with the assistance of TBtools-II [21]. Please detail the exact steps followed to ensure transparency and reproducibility.
3- Lines 350–353: Pfam [https://www.ebi.ac.uk/interpro/ (accessed on 2 June 2024)] and NCBI-CDD [https://www.ncbi.nlm.nih.gov/cdd (accessed on 2 June 2024)] were used to assess all putative non-redundant candidate members. Can you clarify the assessment process followed here?
4- Please verify the tools used in the study and include version numbers where applicable.
5- Figures 1, 2, and 3: These figures could benefit from higher resolution to improve readability.
Author Response
Dear Editor,
Thanks a lot for having reviewed our manuscript (ijms-3320395). We have revised the manuscript and would like to submit it for your consideration. According to your comments and suggestions, we have made corresponding changes. The revisions have been highlighted in the revised manuscript.
I greatly appreciate both your help and that of the referees concerning improvement to this paper. Below you can find point-to-point responses to Reviewers’ comments. We hope that the revised version of the manuscript is now acceptable for publication in your journal.
I look forward to hearing from you soon.
We would like to express our sincere thanks again to you for the constructive and positive comments.
With best wishes,
Yours sincerely,
Yongjie Zhu, Weibiao Liao

Reviewer 2 Report
Comments and Suggestions for Authors
In the current manuscript, the authors studied the MYB gene expression. Although the study seems interesting, the manuscript needs thorough revision. I have listed some comments that may help to improve the manuscript.
1. The abstract needs revision with clear study goals and a brief statement of methods, results, and conclusion.
2. Significant improvement in scientific writing style is needed: “Our samples were extracted with TRIzol reagent (Invitrogen, Carlsbad, CA, USA) to obtain total RNA [51,52].” What do Our samples mean?
3. The authors stated, “This study identified 100 MYB-related genes in R. chinensis, designated as RcMYB001to RcMYB100 (Table 1)” However, it is unclear how the 100 genes were identified and how their physicochemical were obtained. The authors need to organize the results systematically to demonstrate the MYB proteins' characteristics.
4. Figures 1, 2, and 3 are not clear. What is the source of the data in Figure 2?
5. What is the significance of the cis-acting elements in the MYB gene family?
6. The methods section needs revision. The authors stated in the abstract that this study was fully a bioinformatics analysis, but they also performed RNA seq. It is not clear what samples they used for RNA study and how that data was used to draw the conclusions.
7. In the conclusion section, the authors stated they performed tissue-specific expression profiles but did not provide how these data were obtained in Figure 6. Also, how does the tissue-specific gene variation contribute to this study?
8. Figure 7 Investigating the temporal and spatial expression of MYB-related genes in the petals of Rosa chinensis following different phytohormone treatments. What was the rationale for this type of treatment, and how was it performed? The authors need to state this experiment in the methods section.
Author Response

(The authors gave the same response as above.)

Round 2
Reviewer 2 Report
Comments and Suggestions for Authors
The authors successfully answered all my comments in the revised version of the manuscript, which seems much improved.